materials science/spectroscopy

real-time monitoring, dissolution, AA7150, UV–visible spectrophotometer

**Author for correspondence:**
Qingqing Sun
e-mail: sunqq@sustech.edu.cn

# Using real-time UV–visible spectrophotometer to assess an Al–Zn–Mg–Cu alloy's dissolution in acidic solution

Qingqing Sun[1,2], Yang Yang[4], Shuai Wang[2], Panyi Liang[3], Bin Lin[2] and Huabing Li[2]

[1]CAS Key Laboratory of Mechanical Behavior and Design of Materials, Department of Modern Mechanics, University of Science and Technology of China, Hefei, Anhui 230026, People's Republic of China
[2]Department of Mechanical and Energy Engineering, and [3]Department of Chemistry, Southern University of Science and Technology, Shenzhen, Guangdong 518055, People's Republic of China
[4]School of Chemical Engineering, Purdue University, West Lafayette, IN 47907, USA

(iD) QS, 0000-0001-5529-6433

Applicability and limitations of using online non-destructive ultraviolet–visible (UV–vis) spectrophotometer to monitor the dissolution of an Al–Zn–Mg–Cu alloy in HCl-containing solution were studied. Inductively coupled plasma atomic emission spectrometry results indicate that the spectrum absorbance at 252 nm is mainly attributed to Cu-containing complexes. Surprisingly, an hours-long 'induction' period was observed from UV–vis results. This is not a real indicator of induction for Al dissolution as revealed by electrochemical impedance spectrum, actually it reflects the alloy's galvanic corrosion nature that Cu species are released after Al, Zn and Mg species.

## 1. Introduction

The dissolution kinetics of metal alloys is of great importance for corrosion science and engineering [1–4]. Mass loss is a conventional method for corrosion rate assessment that is relatively easy to conduct. However, the data obtained from this method may vary a lot, especially in the early stage of dissolution and for localized corrosion cases, thus lead to a large deviation from the real values. Electrochemical methods like polarization, electrochemical impedance spectrum (EIS) [5] and electrochemical noise (EN) [6] have been introduced and accepted by the corrosion community for decades. Their effectiveness has long been proved. However, the polarization test is destructive and usually imparts polarized potential several hundred mV away from the natural

corrosion potential. As a result, polarization may alter corrosion rate of the studied alloys and could not always reflect the real initiation and propagation process of corrosion under conditions without external disturbance. Even for the so-called non-destructive EIS method, usually it is involved in applying an external sinusoidal potential signal whose amplitude is 5–30 mV with respect to the open circuit potential, thus it should be nearly non-destructive. EN is totally non-destructive and mainly devoted to localized corrosion, however, it could not measure corrosion rate. Microscale and nanoscale techniques like scanning electrochemical microscopy (SECM) [7] and scanning Kelvin probe force microscopy [8] are powerful tools to study localized corrosion mechanism of alloys, but some of these micro/submicro-scale techniques may interfere with the natural corrosion process of alloy, e.g. the feedback mode of SECM [9].

In order to better understand corrosion kinetics under natural conditions without any applied potential, non-destructive methods that do not interfere with the natural corrosion process have been developed. Real-time imaging has been used for non-destructive corrosion assessment [10]. High-magnification imaging from side view disclosed that three different types of hydrogen bubbles were generated during the pure Mg corrosion process. It is proposed that the generation of streams of bubbles at the corrosion front is owing to the disruption of the oxide/hydroxide film that is possible when an anodic current is available and environmental conditions can induce depassivation [10]. However, real-time imaging of bubbles is only qualitative or semi-quantitative. Non-destructive measurement also can be done by measuring the elemental composition of the electrolyte using spectrum methods such as atomic absorption and atomic emission. K. Ogle and his colleagues have applied inductively coupled plasma optical emission spectrometry (ICP-OES) to investigate selective dissolution during the passivation of austenitic stainless steels [11,12], and the anodic and cathodic dissolution of Al and Al–Mg–Cu alloys [13], the formation of zinc oxides during the anodic oxidation of Zn [14], the anodic dissolution of Mg in NaCl and $Na_2SO_4$ electrolytes [15]. Owing to low detecting limit and good performance, ICP-OES is very capable of measuring the concentrations of metal species in soluble solution. For instance, Ogle *et al*. revealed the dissolution rate of Al, Mg and Cu of an Al–Mg–Cu (AA 2024) in 30 g l$^{-1}$ NaCl electrolyte as a function of pH and proposed the corresponding corrosion mechanisms [13].

AA7150 is a typical Al–Zn–Mg–Cu aircraft Al alloy and its corrosion and dissolution, e.g. localized corrosion of second phases and nano-sized precipitations, have been investigated extensively [16–19]. In this work, we attempted to use the *in situ* non-destructive ultraviolet–visible (UV–vis) spectrophotometer method to study dissolution of AA7150 in acid 2 mol l$^{-1}$ NaCl + 0.1 mol l$^{-1}$ HCl medium. The duration of this *in situ* measurement is 24 h. It was of interest to apply this real-time non-destructive technique to the dissolution of Al alloy and see if new insights into this issue might be forthcoming.

# 2. Experimental

## 2.1. Materials

The material used for *in situ* non-destructive UV–vis is a commercially available 7150 aircraft aluminium alloy plate with the thickness of 6 mm and the dimension of $50 \times 50$ mm$^2$, which was received from Alcoa Corp., and treated with T7751 ageing process. The composition is Al-6.5Zn-2.3 Mg-2.3Cu-0.1Mn-0.1Zr-0.04Cr-0.06Ti-0.12Fe-0.12Si, in mass fraction (same for the below context). T3 Cu (purity around 99.7%, as received from Taobao) and Al binary alloys including Al-8Zn, Al-2 Mg, Al-2Cu and Al-2Mn with size $10 \times 10 \times 10$ mm$^3$ were used to do *ex situ* UV–vis absorption measurements. All the Al binary alloys used in this work have been casted, homogenization-treated for 24 h and then hot extruded at 450°C in our laboratory, in order to obtain a homogeneous distribution of composition and structure.

## 2.2. Scanning transmission electron microscopy

Scanning transmission electron microscopy (STEM) specimens were prepared using the focused ion beam *in situ* lift-out technique in an FEI Helios NanoLab (USA) field emission gun dual beam microscope. A voltage of 5 kV was used during the final thinning step to minimize the thickness of the damaged layer created by the ion beam. STEM/high-angle annular dark field images of AA7150 and elements mapping were collected with an FEI/Talos F200X instrument operated at 200 kV.

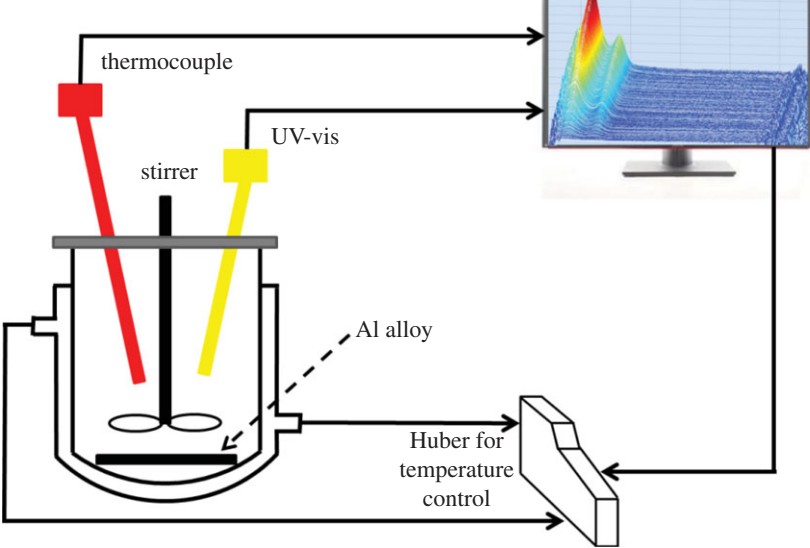

**Figure 1.** Schematic diagram of *in situ* UV–vis monitoring set-up for Al alloy dissolution.

## 2.3. Ultraviolet–visible spectroscopy set-up

Figure 1 presents a schematic diagram of the rig used in the experiments. The corrosion process of 7150 Al alloy was carried out by using 400 ml 2 mol l$^{-1}$ NaCl + 0.1 mol l$^{-1}$ HCl solution in a 500 ml jacketed glass vessel equipped with overhead stirring at 350 r.p.m. (PTFE pitch blade turbine). The temperature was controlled using a PT-100 temperature probe connected to a Huber Ministat 230 thermoregulator. Attenuated total reflectance-UV–vis (Zeiss, MCS621, software PROCESSXPLORER 1.3, Germany) was installed on the vessel. The data from the UV–vis and the Huber is transmitted in real time to the crystallization process informatics system (CRYPRINS) software [20]. In addition, particle vision and measurement (PVM, model V.19, software IC PVM 7.0, Mettler-Toledo) was used to detect the undissolved particles with detection limit down to 0.1 µm. UV–vis is most often used in a quantitative way to determine concentrations of an absorbing species in solution, using the Beer–Lambert law:

$$A = \log_{10}\left(\frac{I_0}{I}\right) = \varepsilon cL, \tag{2.1}$$

where $A$ is the measured absorbance with absorbance units (AU), $I_0$ is the intensity of the incident light for a given wavelength, $I$ is the transmitted intensity, $L$ the path length through the sample and $c$ the concentration of the absorbing species. For each species and wavelength, $\varepsilon$ is a constant known as the molar absorptivity, which is a fundamental molecular property in a given condition.

The background and reference spectra were subtracted immediately after immersing the Al alloy sample into 2 mol l$^{-1}$ NaCl + 0.1 mol l$^{-1}$ HCl solution, thus the change in UV–vis spectra only reflects the information of species newly released into solution. The experiments were performed at 15 and 45°C, respectively.

*Ex situ* UV–vis absorption spectra were taken on an Agilent Technologies Gary 60 (USA) UV–vis spectrophotometer, to determine which kind of complex is responsible to the absorption. AA7150, Al-8Zn, Al-2 Mg, Al-2Cu, Al-2Mn and pure Cu were immersed into 2 mol l$^{-1}$ NaCl + 0.1 mol l$^{-1}$ HCl solution at room temperature for 24 h, then followed by the *ex situ* UV–vis measurements. The corrosive solution (2 mol l$^{-1}$ NaCl + 0.1 mol l$^{-1}$ HCl) without any alloy dissolution into it acts as a reference. At least three parallel experiments were conducted for each solution and the reproductivity was good.

## 2.4. Inductively coupled plasma atomic emission spectrometry

Inductively coupled plasma atomic emission spectrometry (ICP-AES) was used to determine concentrations of dissolved species of alloys after immersion into 2 mol l$^{-1}$ NaCl + 0.1 mol l$^{-1}$ HCl solution for 24 h at room temperature.

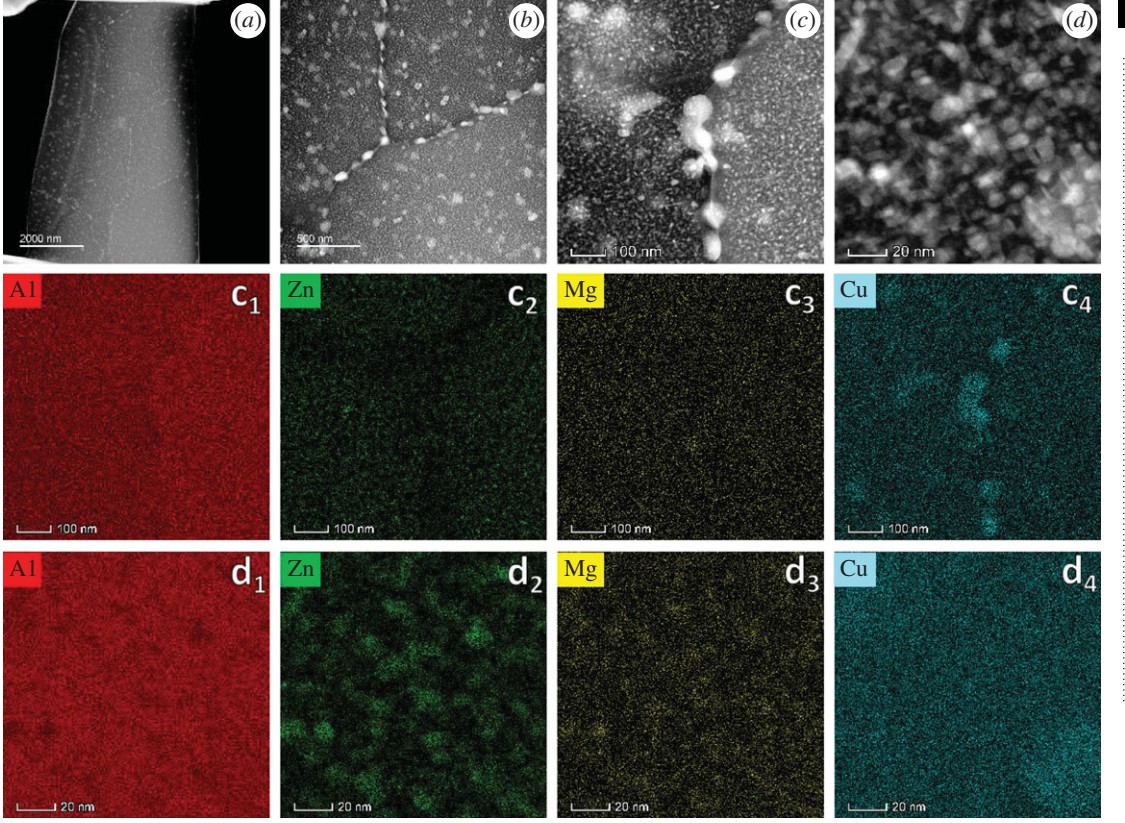

**Figure 2.** HAADF images (*a–d*) of AA7150 and the corresponding elements map ($c_1$–$c_4$ corresponding to figure 2*c*; $d_1$–$d_4$ corresponding to figure 2*d*).

## 2.5. Electrochemical impedance spectrum

A VersaSTAT 3 potentiostat/galvanostat (USA) connected to a three-electrode cell was used for EIS measurements. The working electrode was AA7150 with an immersed area of 1.0 cm$^2$. Platinum gauze and saturated calomel electrode were used as the counter and reference electrodes, respectively. The testing solution is a naturally aerated 2 mol l$^{-1}$ NaCl + 0.1 mol l$^{-1}$ HCl solution, at the temperature of 15°C. EIS measurements were conducted after the open circuit potential (OCP) reached to a stable level, with the frequency ranging from 100 kHz to 1 Hz and the amplitude of the sinusoidal potential signal was 10 mV with respect to the OCP. ZVIEW™ (Scribner Associates Inc., USA) electrochemical analysis software was used to analyse EIS data.

# 3. Results

## 3.1. Scanning transmission electron microscopy

STEM-HAADF images of AA7150 are depicted in figure 2. The grain/sub-grain is about several micros (figure 2*a*) in size, with approximately 50 nm sized copper-riched particles segregating in grain boundaries (figure 2*c*, $c_1$–$c_4$) and 5–10 nm sized Zn, Mg-riched precipitations inside grains (figure 2*d*, $d_1$–$d_4$). Copper-riched particles are also found in the matrix, as shown in the bottom right of figure 2$d_4$.

## 3.2. *In situ* ultraviolet–visible spectroscopy

The UV–vis spectra of AA7150 immersed into 2 mol l$^{-1}$ NaCl + 0.1 mol l$^{-1}$ HCl solution at 15°C and 45°C are, respectively, shown in figure 3*a,b*. Note that the actually measured pH value for the experiment at 15°C is 1.09; while at 45°C it is 0.92. The final pH value after the experiment is 2.9 and 3.2 for 15°C and 45°C, respectively. Corrosion products of Al alloys are very soluble in solution with a pH value below 4, therefore the solution after the experiment is very clear and no particles can be detected

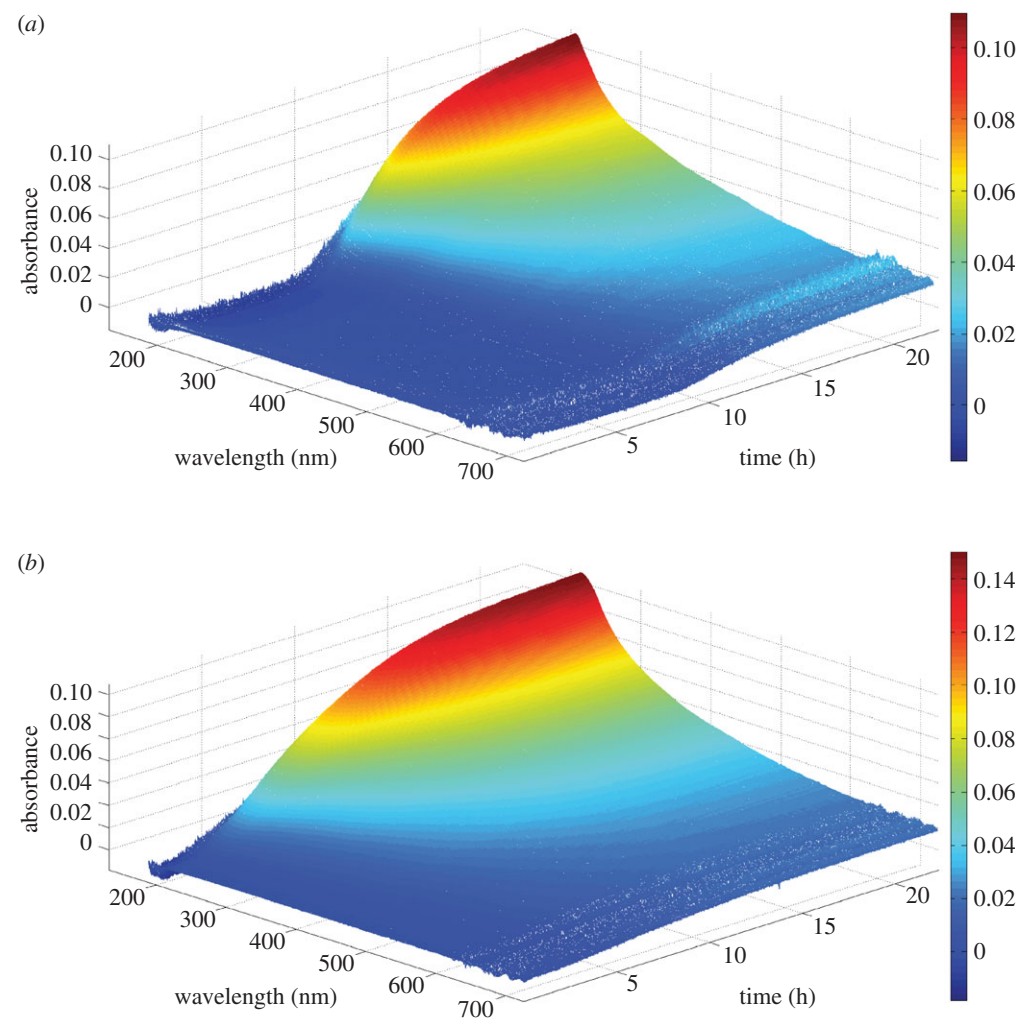

**Figure 3.** *In situ* UV–vis spectra of 7150 Al alloy dissolution as a function of immersion time: (*a*) under 2 mol l$^{-1}$ NaCl + 0.1 mol l$^{-1}$ HCl solution with pH = 1.09 at 15°C; (*b*) under 2 mol l$^{-1}$ NaCl + 0.1 mol l$^{-1}$ HCl solution with pH = 0.92 at 45°C.

using PVM. Thus we believe that the presence of absorbance must be attributed to the accumulation of dissolution species, i.e. Al/Zn/Mg/Cu-containing complexes. However, it is difficult to determine which specific complex/complexes mainly account for the UV–vis absorbance, because inorganic UV–vis absorbance data is quite rare in the literature. Even so, one still can make a point from electron configuration. Compared with Al ($3s^23p^1$) and Mg ($3s^2$), Zn ($3d^{10}4s^2$) and Cu ($3d^{10}4s^1$) more easily form complexes owing the presence of $d$ orbitals. For Zn$^{2+}$ ($3d^{10}$) and Cu$^{2+}$ ($3d^9$), Cu$^{2+}$ has an empty $d$ orbital which makes it more attractive to ligands. Therefore, we hypothesize that the absorbance is mainly owing to the presence of Cu-containing complexes. However, further experiments are needed to validate this hypothesis.

# 4. Discussion

## 4.1. Ultraviolet-visible responding complexes

One can quantitatively analyse UV–vis spectrum according to the Beer–Lambert Law, however, which kind of complex is responsible to the absorbance should be determined first. For the purpose of this, we cast several Al binary alloys based AA7150 main composition (Al-6.5Zn-2.3 Mg-2.3Cu-0.1Mn): Al-8Zn, Al-2 Mg, Al-2Cu and Al-2Mn. Then immersed them into 2 mol l$^{-1}$ NaCl + 0.1 mol l$^{-1}$ HCl solution for 24 h, at room temperature, followed by the *ex situ* UV–vis measurements. Figure 4 shows the UV–vis results (reference and background have been subtracted) of these binary alloys and AA7150. ICP-AES

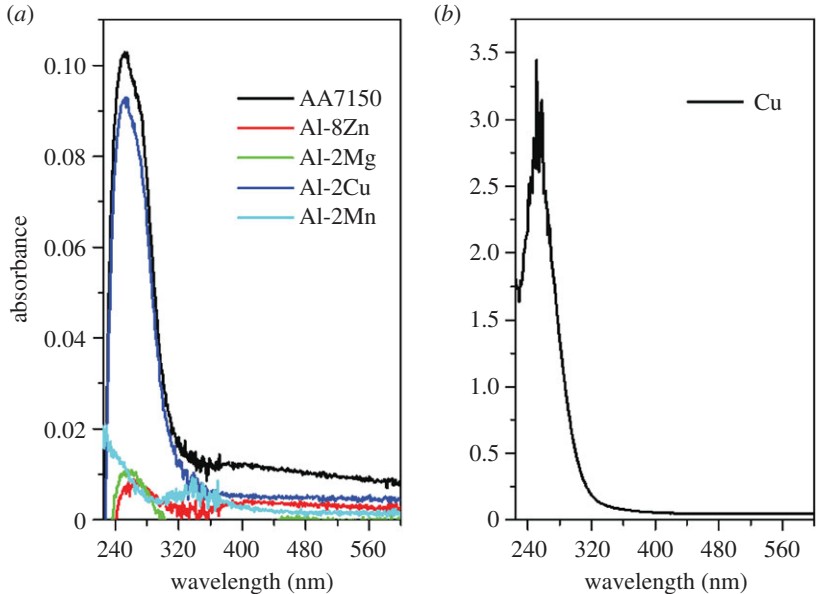

**Figure 4.** *Ex situ* UV–vis spectra of Al binary alloy, AA7150 and pure Cu after immersed into 2 mol l$^{-1}$ NaCl + 0.1 mol l$^{-1}$ HCl solution at room temperature for 24 h: (*a*) Al binary alloys and AA7150; (*b*) pure Cu.

**Table 1.** Compositions of solutions after alloys dissolution for 24 h detected by ICP (mg l$^{-1}$).

|        | Al   | Zn   | Mg   | Cu    | Mn   |
|--------|------|------|------|-------|------|
| AA7150 | 1277 | 3802 | 61.4 | 3.1   | 0    |
| Al-8Zn | 1184 | 5389 | 0.25 | 0     | 0    |
| Al-2Mg | 1032 | 8.9  | 7.5  | 0     | 0    |
| Al-2Cu | 866  | 6.8  | 0.34 | 8.3   | 0    |
| Al-2Mn | 1523 | 1.2  | 0    | 0     | 23.2 |
| Cu     | 0    | 0    | 0    | 154.1 | 0    |

was performed for solutions after the alloys' immersion and the ion concentrations are shown in table 1. At a wavelength of 252 nm, strong absorbance peaks are found for AA7150 and Al-2Cu, with intensity of 0.102 and 0.092, respectively, implying the important role of Cu; while for Al-2 Mg, Al-8Zn and Al-2Mn, the absorbance peak is much less pronounced. The absorbance intensity at the wavelength of 252 nm of Al-2Mg, Al-8Zn and Al-2Mn is 0.011, 0.009 and 0.012, respectively, which are very close to each other. For Al-2Mn, a flat peak at 340 nm with absorbance intensity of 0.008 also can be observed, which should be owing to Mn-containing complexes. As expected, the peak at 340 nm is absent for AA7150 owing to its low content of Mn. As can be seen from table 1, the concentration of Al is quite comparable for Al-2 Mg, Al-8Zn and Al-2Mn alloys, which is the same with absorbance values at 252 nm. Therefore, it is reasonable to believe that for Al-2Mg, Al-8Zn and Al-2Mn, the absorbance at 252 nm is mainly owing to Al-containing complexes. Comparing ion concentrations and UV–vis results of Al-8Zn and AA7150, it is easy to find that the effect of Zn on absorbance is negligible. Readers probably question why the absorbance at 252 nm of Al-2Mg is not mainly owing to the Mg ion, as for Al-2Mg and AA7150, the Mg concentration increases from 7.5 to 61.4 mg l$^{-1}$ and the absorbance increases from 0.011 to 0.102. According to Mokaddem *et al.*'s work [13], Mg and Al ions are released into solution immediately during dissolution. Thus, if Mg-containing complexes play a major role in absorbance for Al-2Mg and AA7150, then the absorbance curve depicted in figure 3*a* should have a gradual increasing trend, instead of an abrupt turning point at 7.5 h. Therefore, for Al-2Mg alloy it is Al-containing complexes, instead of Mg species, that mainly accounted for the UV–vis response. We further conducted *ex situ* UV–vis measurements for T3 pure Cu, and the presence of an ultra-strong absorbance peak at 252 nm shown in figure 4*b* strongly indicates Cu's contribution. In addition, comparing absorbance and concentration information of 7150 Al alloy, Al-2Cu and other Al binary alloys, Cu's contribution on absorbance is much

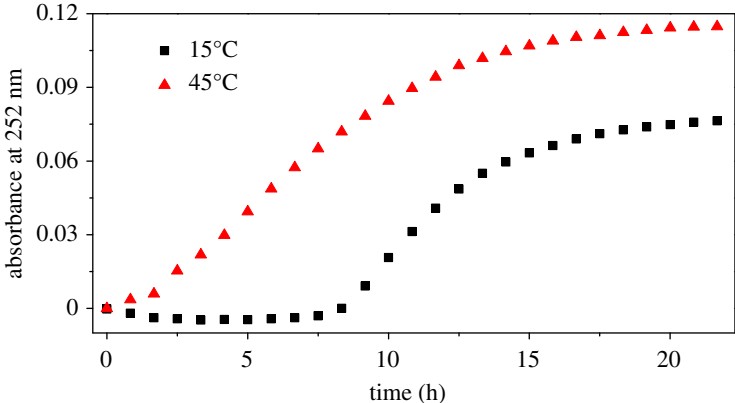

**Figure 5.** UV–vis absorbance at 252 nm for 7150 Al alloy dissolution as a function of immersion time: under 2 mol l$^{-1}$ NaCl + 0.1 mol l$^{-1}$ HCl solution with pH = 1.09, $T$ = 15°C; under 2 mol l$^{-1}$ NaCl + 0.1 mol l$^{-1}$ HCl solution with pH = 0.92, $T$ = 45°C. An 'induction' period of UV–vis absorbance can be easily seen from this curve.

stronger than that of Al. Therefore, based on the above analysis, the absorbance at 252 nm is mainly accounted for on copper. Another thing worth noting is that the concentration of Cu is not simply proportional to absorbance intensity. For instance, after subtracting the contribution of Al (approx. 0.01), Cu absorbance of pure Cu, Al-2Cu and AA7150, respectively, is 3.4, 0.08 and 0.09, while the respective Cu concentration is 154.1, 8.3 and 3.1 mg l$^{-1}$. This probably is because the formation and solution structures of Cu-containing complexes are affected by the presence of other species like Al, Mg and Zn ions.

## 4.2. S-shape curve and 'induction' period

The UV–vis absorbance at the wavelength of 252 nm as a function of immersion time is shown in figure 5, and both 15°C and 45°C specimens show S-shape curves. At the early stage, the curve is quite flat, which seems like an 'induction' period. For immersion performed at 15°C, the 'induction' period is as long as 7.5 h; at 45°C it is about 1.5 h. Slightly higher H$^+$ concentration may play a role in the much shorter 'induction' period at 45°C, but higher temperature is believed to be the main reason. After the 'induction' period, corrosion rate was found increasing dramatically to the peak value, followed by a gradual drop owing to the consumption of acid. For samples at 15°C, the final platform of absorbance is approximately 0.075. For samples at 45°C, the final platform of absorbance is approximately 0.117. The difference of final platform value might be owing to the difference in initial pH value. The initial pH value of solution at 15°C is 1.09, which is a little bit higher than that of solution at 45°C (0.92).

The studied solution is 2 mol l$^{-1}$ NaCl + 0.1 mol l$^{-1}$ HCl with pH around 1, which is quite acidic. Thus, the evidence of hours-long 'induction' period of the dissolution of AA7150 in such severe circumstance is beyond our expectation. If this induction is true, one may demonstrate that the persistence of passive film of Al alloys in 2 mol l$^{-1}$ NaCl + 0.1 mol l$^{-1}$ HCl solution could be much stronger than we thought. However, this is contradictory with Mokaddem *et al.*'s work [13], in which the dissolution rate of Al, Mg and Cu of an Al–Mg–Cu (AA 2024) in 30 g l$^{-1}$ NaCl electrolyte under open circuit potential as a function of pH was revealed using non-destructive ICP-OES, and the induction period was not found in his work. In addition, the EIS results also indicate the hours-long induction period is absent for AA7150 in 2 mol l$^{-1}$ NaCl + 0.1 mol l$^{-1}$ HCl solution. EIS results of AA7150 in the same studied solution at 15°C as a function of immersion time are shown in figure 6. An equivalent electrical circuit model inserted in figure 6*d* is used for quantitative analysis. The physical meaning of the equivalent circuit elements is as follows: $R_s$ is the ohmic resistance of the electrolyte, CPE is the constant phase element of double-layer capacitance and $R_{ct}$ is charge transfer resistance. The element CPE is used to represent the possibility of a non-ideal capacitance. $R_{ct}$ evolution shown in figure 6*d* declined dramatically in the first 1.25 h and then decreased slowly to a minimal value at 10 h. With further immersion time, the value of $R_{ct}$ increased slightly and was followed by a flat platform after 24 h' immersion. It is explicit that no induction period was found from EIS measurements.

Here, we propose that the 'induction' period shown in UV–vis curve is owing to the alloy's galvanic corrosion nature. The schematic diagram showing the evolution of UV–vis absorbance at 252 nm is illustrated in figure 7. At the 'induction' stage, only Al/Zn/Mg ions were released into solution. This is because for Al–Zn–Mg–Cu alloys, anodic phases (Al matrix, MgZn$_2$) are preferentially dissolved [16].

**Figure 6.** EIS of 7150 Al alloy in 2 mol l$^{-1}$ NaCl + 0.1 mol l$^{-1}$ HCl solution at 15°C as a function of immersion time: (*a*) Nyquist plots; (*b,c*) Bode plots; (*d*) charge transfer resistance $R_{ct}$.

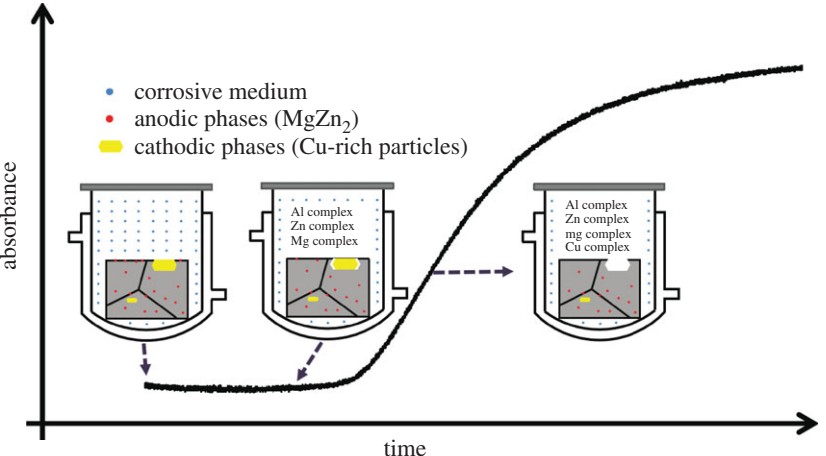

**Figure 7.** Schematic diagram showing the evolution of UV–vis absorbance at the wavelength of 252 nm.

However, owing to their electron configurations, Al/Zn/Mg ions are hard to form complexes which can be detected by UV–vis (figure 4*a*). Thus, an 'induction' period is shown in absorbance curve. Cu-containing cathodic particles, such as Al$_2$Cu and Al$_2$CuMg, would separate with the Al matrix after a certain period owing to galvanic corrosion and strong stirring of the solution [16,18]. The Cu-rich particles exfoliated into acidic solution, dissolved and formed UV–vis detectable Cu-containing complexes (figure 4). Thus, UV–vis absorbance is shown after several hours of immersion, and increases with the accumulation of Cu-containing complexes.

# 5. Conclusion

Real-time monitoring of an Al–Zn–Mg–Cu alloy dissolution process in 2 mol l$^{-1}$ NaCl + 0.1 mol l$^{-1}$ HCl solution using a UV–vis spectrophotometer was studied. UV–vis absorbance at 252 nm is mainly owing to Cu-containing complexes formed during the dissolution process. An 'induction' period lasting several hours was observed from the UV–vis spectrum. However, that is not a true indicator of the presence of an induction period for Al alloy dissolution. Actually, this is owing to 7150 Al alloy's localized galvanic corrosion nature: Cu species are released after Al, Zn or Mg species. Though non-destructive, the UV–vis method is limited for alloy dissolution rate assessment because it only reflects information of metal ion complexes that can be detected by spectrum.

Data accessibility. All the datasets supporting this article are available at the Dryad Digital Repository (https://dx.doi.org/10.5061/dryad.pzgmsbch2) [21].

Authors' contributions. Q.S. and Y.Y. initiated and conducted the experiments on real-time UV-vis, P.L. performed the offline UV-vis part. Q.S., H.L. and B.L. wrote the manuscript. S.W. revised the manuscript.

Competing interests. The authors declare that they have no conflict of interest.

Funding. This work was supported by Purdue Center for Materials Processing Research at Purdue University, Start-up fund of Sustech and China Scholarship Council. Project funding by China Postdoctoral Science Foundation (223278) and Youth Innovation Project funding by Education Department of Guangdong Province (2018KQNCX227) is also gratefully acknowledged.

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
