## [Reviewer comments · Royal Society Open Science]

Review History

RSOS-200461.R0 (Original submission)

Review form: Reviewer 1

Is the manuscript scientifically sound in its present form?

Yes

Are the interpretations and conclusions justified by the results?

Yes

Is the language acceptable?

Yes

Do you have any ethical concerns with this paper?

No

Have you any concerns about statistical analyses in this paper?

No

Recommendation?

Accept with minor revision (please list in comments)

Comments to the Author(s)

This work deserves publication in this journal as it brings an interesting way to investigate corrosion and dissolution of metals. Certainly related strategies have been used or are in use today, some of them referred in this work, however the results are sufficiently different to justify publication in this journal.

Though, the following points must be taken into consideration in the necessary revision.

References 1-5 seem not appropriate to support the great importance of metal dissolution in corrosion or to give further information about the importance of corrosion science and engineering. The criteria for choosing these references, if any, is not clear. It seems that any corrosion paper could have been chosen. Books, Handbooks, Review papers or Critical Perspectives are better references.

The previous comment is particularly applicable to references 5, 6 and 7. The references for techniques like polarization, EIS and Noise, should be either dedicated books, review papers or the first papers where each technique has appeared.

Page 2, line 19. The comment that EIS accelerates corrosion is incorrect. This might occur if the technique is not correctly performed, or performed in circumstances where causality, linearity and stability (the three primary conditions that must be met during an EIS experiment for the results be considered valid) are not obeyed. If this happens it cannot be called an EIS experiment.

Page 3, lines 23-27. It is also incorrect to say that microscale and nanoscale techniques are destructive. They just follow the natural corrosion process, which is by itself destructive due to its very nature. The techniques do not interfere with the process. The same applies to the UV-Vis spectrometry technique presented in this work.

Page 3, lines 30-31. These lines make no sense. The claim that non-destructive methods need to be developed implies that they do not exist. They do! Plenty of methods exist, including EIS, SECM, SKPFM, optical monitoring, etc.

A clear difference must be stated between the form of inducing degradation (which can be natural or can be forced, with polarization for example) and the methods to analyse the degradation (which can be destructive or non-destructive). In any case, the natural process of corrosion is destructive. The approach proposed by the authors is a spectroscopic method (non-destructive) with the dissolution of an alloy in acid (destructive). This must be clarified to avoid confusion and misinterpretations.

Page 2, last line. The 10 minutes are presented here as a limitation. No. If necessary the measurements can be continued for any time desired.

Page 3, line 15. Which debate are authors referring to?

2.1. Materials

What is T3 Cu?

From where did authors obtained the Al binary alloys? Sizes and shapes? The compositions were uniform and homogeneous along all the volume of the samples?

2.2. STEM

What was the main voltage used with the field emission gun?

The acronym HAADF must be introduced.

2.3. UV-vis

Line 49. Revise "...the solution was carried out in a (...) glass vessel..."

The caption of Fig 1 does not need to state the experimental details that are already written in the experimental section.

Page 4, line 49. It is referred that mass loss tests were performed, but there is no information about it in the experimental section. Measurements after what time of immersion?

Page 6, line 44. Introduce the apparatus used for the PVM.

Table 1. Include in the caption the time of immersion.

Page 10, line 22. The CPE is related to the passive films, as is written, or to the double layer capacitance?

Figure 6. The Nyquist plot in a) must have orthonormal axes (same length for the same impedance in both axes).

The legend of Y axis in c) is strange. Why the minus signal? And it should be " $\log |Z|$ ".

Page 11, first line. "fake" should be replaced by a better word.

Figure 7. It is better to put axes in the graph. The colour legend indicates "paicles" instead of particles. Consider changing the colour dots in the image: blue for solution and grey for the alloy. Seems more common-sense.

Conclusions

The "non-destructive" before UV-Vis spectrophotometer is unnecessary. Because it is obvious the technique does not interfere with the metal degradation, and also because it may lead to a wrong conclusion: the metal corrodes and is destroyed in a natural manner but the way it is written it seems that at the end of the experiment the sample remains pristine, unspoiled.

"due to the empty d orbital of Cu^{2+} " is not a conclusion from this work. It should be deleted without any prejudice to the message.

Review the last 4 words, "complexes that spectrum detectable."

References 6 and 17 are the same.

Review form: Reviewer 2

Is the manuscript scientifically sound in its present form?

No

Are the interpretations and conclusions justified by the results?

No

Is the language acceptable?

Yes

Do you have any ethical concerns with this paper?

No

Have you any concerns about statistical analyses in this paper?

Yes

Recommendation?

Accept with minor revision (please list in comments)

Comments to the Author(s)

While the manuscript is interesting and describes the possible uses and limitations of UV-vis for in-situ metal dissolution, there is one main question issue that needs to be clarified and some minor comments.

While very well written and easy to understand there are a few minor errors that could be fixed with a proof read.

In the introduction there is a sentence : Mass loss is a good and conventional..... The word good is very vague and should just be removed, so it reads something like: Mass loss is a conventional method for corrosion rate.....that is relatively easy to conduct.

Also in the introduction it says that polarization polarises to several hundred millivolts. Is the point here that polarisation is destructive or that it is not a reliable measure of corrosion current? While it is destructive, the reliability of a polarisation test should always be confirmed by conducting separate tests for the anodic and cathodic arm that should match the full scan. If this is done then it is a reliable method.

Need references for the statement that EIS accelerates corrosion p2, line 19.

Final line of the introduction states that: It was of interest to apply this real-time non-destructive technique to the dissolution of Al alloy and see if new insights into this debate might be forthcoming. A debate means that there are opposing views as to what is occurring, what are they? Or is this technique just making the processes that occur clearer?

The results section is clear and well written, however there is one major issue that needs to be cleared up. Figure 4a and Table 1 shows that for a similar level of Cu in solution for Al-2Cu (8.3mg/L) and Mg for Al-2Mg (7.5mg/L) the UV-vis shows a roughly 4 times higher Cu signal (intensity 0.092 for Cu to 0.026 for Mg). However, for the AA7150, Table 1 shows around 20 times as much Mg as Cu present in solution, but this is ignored as the author states that the absorbance at 252nm is mainly accounted for by copper. Is the UV-vis intensity not proportional to concentration? Please explain how the signal for Mg in the AA7150 solution can largely be ignored when there is 20 times more in solution and its signal was shown to be roughly 4 times smaller than that of Cu for a comparable amount in solution.

Decision letter (RSOS-200461.R0)

Dear Dr Sun:

Title: Using real-time UV-visible spectrophotometer to assess an Al-Zn-Mg-Cu alloy's dissolution in acidic solution

Manuscript ID: RSOS-200461

Thank you for submitting the above manuscript to Royal Society Open Science. On behalf of the Editors and the Royal Society of Chemistry, I am pleased to inform you that your manuscript will be accepted for publication in Royal Society Open Science subject to minor revision in accordance with the referee suggestions. Please find the reviewers' comments at the end of this email.

The reviewers and handling editors have recommended publication, but also suggest some minor revisions to your manuscript. Therefore, I invite you to respond to the comments and revise your manuscript.

Because the schedule for publication is very tight, it is a condition of publication that you submit the revised version of your manuscript before 19-Jul-2020. Please note that the revision deadline will expire at 00.00am on this date. If you do not think you will be able to meet this date please let me know immediately.

Kind regards,
Dr Laura Smith
Publishing Editor, Journals

On behalf of the Subject Editor Professor Anthony Stace and the Associate Editor Dr Dattatray Late.

RSC Subject Editor:
Comments to the Author:
(There are no comments.)

RSC Associate Editor:
Comments to the Author:
1) Introduction need more elaboration highlighting importance of the work and novelty etc.
2) XPS data and analysis need to be added.

Reviewer comments to Author:
Reviewer: 1

Comments to the Author(s)
This work deserves publication in this journal as it brings an interesting way to investigate corrosion and dissolution of metals. Certainly related strategies have been used or are in use today, some of them referred in this work, however the results are sufficiently different to justify publication in this journal.

Though, the following points must be taken into consideration in the necessary revision.

References 1-5 seem not appropriate to support the great importance of metal dissolution in corrosion or to give further information about the importance of corrosion science and engineering. The criteria for choosing these references, if any, is not clear. It seems that any corrosion paper could have been chosen. Books, Handbooks, Review papers or Critical Perspectives are better references.

The previous comment is particularly applicable to references 5, 6 and 7. The references for techniques like polarization, EIS and Noise, should be either dedicated books, review papers or the first papers where each technique has appeared.

Page 2, line 19. The comment that EIS accelerates corrosion is incorrect. This might occur if the technique is not correctly performed, or performed in circumstances where causality, linearity and stability (the three primary conditions that must be met during an EIS experiment for the results be considered valid) are not obeyed. If this happens it cannot be called an EIS experiment.

Page 3, lines 23-27. It is also incorrect to say that microscale and nanoscale techniques are destructive. They just follow the natural corrosion process, which is by itself destructive due to its very nature. The techniques do not interfere with the process. The same applies to the UV-Vis spectrometry technique presented in this work.

Page 3, lines 30-31. These lines make no sense. The claim that non-destructive methods need to be developed implies that they do not exist. They do! Plenty of methods exist, including EIS, SECM, SKPFM, optical monitoring, etc.

A clear difference must be stated between the form of inducing degradation (which can be natural or can be forced, with polarization for example) and the methods to analyse the degradation (which can be destructive or non-destructive). In any case, the natural process of corrosion is destructive. The approach proposed by the authors is a spectroscopic method (non-

destructive) with the dissolution of an alloy in acid (destructive). This must be clarified to avoid confusion and misinterpretations.

Page 2, last line. The 10 minutes are presented here as a limitation. No. If necessary the measurements can be continued for any time desired.

Page 3, line 15. Which debate are authors referring to?

2.1. Materials

What is T3 Cu?

From where did authors obtained the Al binary alloys? Sizes and shapes? The compositions were uniform and homogeneous along all the volume of the samples?

2.2. STEM

What was the main voltage used with the field emission gun?

The acronym HAADF must be introduced.

2.3. UV-vis

Line 49. Revise "...the solution was carried out in a (...) glass vessel..."

The caption of Fig 1 does not need to state the experimental details that are already written in the experimental section.

Page 4, line 49. It is referred that mass loss tests were performed, but there is no information about it in the experimental section. Measurements after what time of immersion?

Page 6, line 44. Introduce the apparatus used for the PVM.

Table 1. Include in the caption the time of immersion.

Page 10, line 22. The CPE is related to the passive films, as is written, or to the double layer capacitance?

Figure 6. The Nyquist plot in a) must have orthonormal axes (same length for the same impedance in both axes).

The legend of Y axis in c) is strange. Why the minus signal? And it should be " $\log |Z|$ ".

Page 11, first line. "fake" should be replaced by a better word.

Figure 7. It is better to put axes in the graph. The colour legend indicates "paicles" instead of particles. Consider changing the colour dots in the image: blue for solution and grey for the alloy. Seems more common-sense.

Conclusions

The "non-destructive" before UV-Vis spectrophotometer is unnecessary. Because it is obvious the technique does not interfere with the metal degradation, and also because it may lead to a wrong conclusion: the metal corrodes and is destroyed in a natural manner but the way it is written it seems that at the end of the experiment the sample remains pristine, unspoiled.

"due to the empty d orbital of Cu^{2+} " is not a conclusion from this work. It should be deleted without any prejudice to the message.

Review the last 4 words, "complexes that spectrum detectable."

References 6 and 17 are the same.

Reviewer: 2

Comments to the Author(s)

While the manuscript is interesting and describes the possible uses and limitations of UV-vis for in-situ metal dissolution, there is one main question issue that needs to be clarified and some minor comments.

While very well written and easy to understand there are a few minor errors that could be fixed with a proof read.

In the introduction there is a sentence : Mass loss is a good and conventional..... The word good is very vague and should just be removed, so it reads something like: Mass loss is a conventional method for corrosion rate.....that is relatively easy to conduct.

Also in the introduction it says that polarization polarises to several hundred millivolts. Is the point here that polarisation is destructive or that it is not a reliable measure of corrosion current? While it is destructive, the reliability of a polarisation test should always be confirmed by conducting separate tests for the anodic and cathodic arm that should match the full scan. If this is done then it is a reliable method.

Need references for the statement that EIS accelerates corrosion p2, line 19.

Final line of the introduction states that: It was of interest to apply this real-time non-destructive technique to the dissolution of Al alloy and see if new insights into this debate might be forthcoming. A debate means that there are opposing views as to what is occurring, what are they? Or is this technique just making the processes that occur clearer?

The results section is clear and well written, however there is one major issue that needs to be cleared up. Figure 4a and Table 1 shows that for a similar level of Cu in solution for Al-2Cu (8.3mg/L) and Mg for Al-2Mg (7.5mg/L) the UV-vis shows a roughly 4 times higher Cu signal (intensity 0.092 for Cu to 0.026 for Mg). However, for the AA7150, Table 1 shows around 20 times as much Mg as Cu present in solution, but this is ignored as the author states that the absorbance at 252nm is mainly accounted for by copper. Is the UV-vis intensity not proportional to concentration? Please explain how the signal for Mg in the AA7150 solution can largely be ignored when there is 20 times more in solution and its signal was shown to be roughly 4 times smaller than that of Cu for a comparable amount in solution.

Author's Response to Decision Letter for (RSOS-200461.R0)

See Appendix A.

RSOS-200461.R1 (Revision)

Review form: Reviewer 1

Is the manuscript scientifically sound in its present form?

Yes

Are the interpretations and conclusions justified by the results?

Yes

Is the language acceptable?

Yes

Do you have any ethical concerns with this paper?

No

Have you any concerns about statistical analyses in this paper?

No

Recommendation?

Accept with minor revision (please list in comments)

Comments to the Author(s)

This reviewer is satisfied with the changes made by the authors.

Just pay attention to the following:

Kelvin is a name so in page 2 line 7 write "scanning Kelvin probe force microscopy"

Page 2 lines 9-11. The phrase is more correct if written as "Some of these micro/submicro-scale techniques may sometimes interfere with the natural corrosion process...".

Contrary to what this introduction tries to state most localised techniques do not interfere with the corrosion process: LEIS, SKP, SKPFM, SVET, even SECM (because in SECM corrosion studies it is pH, O₂ or metal ions reduction - after the natural oxidation by the corrosive process - that are measured, without any external polarization).

Give the source/producer of the binary alloys (Al-8Zn, Al-2Mg, Al-2Cu, Al-2Mn).

Always state the country of each equipment manufacturer.

EIS. Fig. 6. Only 1 time constant?

Review form: Reviewer 2

Is the manuscript scientifically sound in its present form?

No

Are the interpretations and conclusions justified by the results?

No

Is the language acceptable?

Yes

Do you have any ethical concerns with this paper?

No

Have you any concerns about statistical analyses in this paper?

No

Recommendation?

Major revision is needed (please make suggestions in comments)

Comments to the Author(s)

The authors have addressed most of the reviewers comments well, however a question remains as to how much the Mg in AA7150 contributes to the UV-vis signal. While it is clear that Cu gives

a stronger signal than Mg, there is much more Mg present in the alloy, however the authors choose to ignore its contribution. This may be true, but proof is needed either from a reference or through some experiments in which the amounts of Cu and Mg together in a solution are varied and the effect on the UV-vis signal is determined. The authors themselves state: '... the formation of absorbance sensitive Mg complexes is suppressed in the presence of Cu ions, and probably Cu-Mg-containing complexes formed, but these hypotheses still need further proofs.' Since this is the main point of the manuscript then it needs to be proven.

Decision letter (RSOS-200461.R1)

Dear Dr Sun:

Title: Using real-time UV-visible spectrophotometer to assess an Al-Zn-Mg-Cu alloy's dissolution in acidic solution
Manuscript ID: RSOS-200461.R1

Thank you for submitting the above manuscript to Royal Society Open Science. On behalf of the Editors and the Royal Society of Chemistry, I am pleased to inform you that your manuscript will be accepted for publication in Royal Society Open Science subject to minor revision in accordance with the referee suggestions. Please find the reviewers' comments at the end of this email.

The reviewers and handling editors have recommended publication, but also suggest some minor revisions to your manuscript. Therefore, I invite you to respond to the comments and revise your manuscript.

Because the schedule for publication is very tight, it is a condition of publication that you submit the revised version of your manuscript before 08-Aug-2020. Please note that the revision deadline will expire at 00.00am on this date. If you do not think you will be able to meet this date please let me know immediately.

- 1) A text file of the manuscript (tex, txt, rtf, docx or doc), references, tables (including captions) and figure captions. Do not upload a PDF as your "Main Document".
- 2) A separate electronic file of each figure (EPS or print-quality PDF preferred (either format should be produced directly from original creation package), or original software format)

- 3) Included a 100 word media summary of your paper when requested at submission. Please ensure you have entered correct contact details (email, institution and telephone) in your user account
- 4) Included the raw data to support the claims made in your paper. You can either include your data as electronic supplementary material or upload to a repository and include the relevant doi within your manuscript
- 5) All supplementary materials accompanying an accepted article will be treated as in their final form. Note that the Royal Society will neither edit nor typeset supplementary material and it will be hosted as provided. Please ensure that the supplementary material includes the paper details where possible (authors, article title, journal name).

Kind regards,
Dr Laura Smith
Publishing Editor, Journals

On behalf of the Subject Editor Professor Anthony Stace and the Associate Editor Dr Dattatray Late.

RSC Associate Editor:
Comments to the Author:
(There are no comments.)

RSC Subject Editor:
Comments to the Author:
(There are no comments.)

Reviewer comments to Author:
Reviewer: 2

Comments to the Author(s)
The authors have addressed most of the reviewers comments well, however a question remains as to how much the Mg in AA7150 contributes to the UV-vis signal. While it is clear that Cu gives a stronger signal than Mg, there is much more Mg present in the alloy, however the authors choose to ignore its contribution. This may be true, but proof is needed either from a reference or

through some experiments in which the amounts of Cu and Mg together in a solution are varied and the effect on the UV-vis signal is determined. The authors themselves state: '... the formation of absorbance sensitive Mg complexes is suppressed in the presence of Cu ions, and probably Cu-Mg-containing complexes formed, but these hypotheses still need further proofs.' Since this is the main point of the manuscript then it needs to be proven.

Reviewer: 1

Comments to the Author(s)

This reviewer is satisfied with the changes made by the authors.

Just pay attention to the following:

Kelvin is a name so in page 2 line 7 write "scanning Kelvin probe force microscopy"

Page 2 lines 9-11. The phrase is more correct if written as "Some of these micro/submicro-scale techniques may sometimes interfere with the natural corrosion process...".

Contrary to what this introduction tries to state most localised techniques do not interfere with the corrosion process: LEIS, SKP, SKPFM, SVET, even SECM (because in SECM corrosion studies it is pH, O₂ or metal ions reduction - after the natural oxidation by the corrosive process - that are measured, without any external polarization).

Give the source/producer of the binary alloys (Al-8Zn, Al-2Mg, Al-2Cu, Al-2Mn).

Always state the country of each equipment manufacturer.

EIS. Fig. 6. Only 1 time constant?

Author's Response to Decision Letter for (RSOS-200461.R1)

See Appendix B.

RSOS-200461.R2 (Revision)

Review form: Reviewer 1

Is the manuscript scientifically sound in its present form?

Yes

Are the interpretations and conclusions justified by the results?

Yes

Is the language acceptable?

Yes

Do you have any ethical concerns with this paper?

No

Have you any concerns about statistical analyses in this paper?

No

Recommendation?

Accept with minor revision (please list in comments)

Comments to the Author(s)

Just two small points needing revision.

1) But some of these micro/submicro-scale techniques MAY/MIGHT interfere with natural corrosion process...

2) In the experimental part you should say that the binary alloys were produced by yourselves (and give some small details about how they were prepared).

Review form: Reviewer 2**Is the manuscript scientifically sound in its present form?**

Yes

Are the interpretations and conclusions justified by the results?

No

Is the language acceptable?

Yes

Do you have any ethical concerns with this paper?

No

Have you any concerns about statistical analyses in this paper?

Yes

Recommendation?

Accept with minor revision (please list in comments)

Comments to the Author(s)

While the authors have made a valid change to the data presented that explains why they believe Cu is the major contributor to the UV-Vis peak in the AA7150 solution, they have not changed the peak reading in their discussion and still refer to a peak height for the Al-2Mg solution of 0.026 (p10 line 7) when the new curve shows a value closer to 0.01. Additionally, they seem to have chosen the peak height of just one of their repeats (the number of repeats was not mentioned in the experimental section) and offer no average values or error analysis. While this new value of the peak height is now close to those of Zn and Mn, which the authors state are so low they can be ignored, a more thorough explanation on why these can be ignored should be supplied because, unlike for Mn and Zn, the ICP results in Table 1 show that the amount of Mg present in solution for the AA7150 (61.4 mg/L) is much increased as compared to the Al-2Mg solution (7.5) from which the calibration for Mg is measured, so if there is any contribution from Mg in the UV-Vis peak it will be much increased in the AA7150 measurement. This is also important because the UV-Vis peak height in Figure 4 for the AA7150 solution actually increases compared to that of the Al-2Cu solution, even though Table 1 shows less than half the amount of Cu in solution for the AA7150 solution (3.1 vs 8.3 for the Al-2Cu solution). If the authors can explain in the manuscript why a peak height of around 0.1 on the UV-Vis can be ignored then the paper should be accepted.

Decision letter (RSOS-200461.R2)

Dear Dr Sun:

Title: Using real-time UV-visible spectrophotometer to assess an Al-Zn-Mg-Cu alloy's dissolution in acidic solution
Manuscript ID: RSOS-200461.R2

Thank you for submitting the above manuscript to Royal Society Open Science. On behalf of the Editors and the Royal Society of Chemistry, I am pleased to inform you that your manuscript will be accepted for publication in Royal Society Open Science subject to minor revision in accordance with the referee suggestions. Please find the reviewers' comments at the end of this email.

The reviewers and handling editors have recommended publication, but also suggest some minor revisions to your manuscript. Therefore, I invite you to respond to the comments and revise your manuscript.

Because the schedule for publication is very tight, it is a condition of publication that you submit the revised version of your manuscript before 02-Sep-2020. Please note that the revision deadline will expire at 00.00am on this date. If you do not think you will be able to meet this date please let me know immediately.

Kind regards,
Dr Laura Smith
Publishing Editor, Journals

On behalf of the Subject Editor Professor Anthony Stace and the Associate Editor Dr Dattatray Late.

RSC Associate Editor:
Comments to the Author:
Accept with minor revisions

RSC Subject Editor:
Comments to the Author:
(There are no comments.)

Reviewer comments to Author:
Reviewer: 1
Comments to the Author(s)
Just two small points needing revision.

1) But some of these micro/submicro-scale techniques MAY/MIGHT interfere with natural corrosion process...

2) In the experimental part you should say that the binary alloys were produced by yourselves (and give some small details about how they were prepared).

Reviewer: 2

Comments to the Author(s)
While the authors have made a valid change to the data presented that explains why they believe Cu is the major contributor to the UV-Vis peak in the AA7150 solution, they have not changed the peak reading in their discussion and still refer to a peak height for the Al-2Mg solution of 0.026 (p10 line 7) when the new curve shows a value closer to 0.01. Additionally, they seem to have

chosen the peak height of just one of their repeats (the number of repeats was not mentioned in the experimental section) and offer no average values or error analysis. While this new value of the peak height is now close to those of Zn and Mn, which the authors state are so low they can be ignored, a more thorough explanation on why these can be ignored should be supplied because, unlike for Mn and Zn, the ICP results in Table 1 show that the amount of Mg present in solution for the AA7150 (61.4 mg/L) is much increased as compared to the Al-2Mg solution (7.5) from which the calibration for Mg is measured, so if there is any contribution from Mg in the UV-Vis peak it will be much increased in the AA7150 measurement. This is also important because the UV-Vis peak height in Figure 4 for the AA7150 solution actually increases compared to that of the Al-2Cu solution, even though Table 1 shows less than half the amount of Cu in solution for the AA7150 solution (3.1 vs 8.3 for the Al-2Cu solution). If the authors can explain in the manuscript why a peak height of around 0.1 on the UV-Vis can be ignored then the paper should be accepted.

Author's Response to Decision Letter for (RSOS-200461.R2)

See Appendix C.

RSOS-200461.R3 (Revision)

Review form: Reviewer 1

Is the manuscript scientifically sound in its present form?

Yes

Are the interpretations and conclusions justified by the results?

Yes

Is the language acceptable?

Yes

Do you have any ethical concerns with this paper?

No

Have you any concerns about statistical analyses in this paper?

No

Recommendation?

Accept as is

Comments to the Author(s)

The manuscript is acceptable for publication as is.

Review form: Reviewer 2

Is the manuscript scientifically sound in its present form?

Yes

Are the interpretations and conclusions justified by the results?

Yes

Is the language acceptable?

Yes

Do you have any ethical concerns with this paper?

No

Have you any concerns about statistical analyses in this paper?

No

Recommendation?

Accept as is

Comments to the Author(s)

The addition the authors have put in perfectly addresses the concerns about the assignment of the peaks and should be published with no further changes required.

Decision letter (RSOS-200461.R3)

Dear Dr Sun:

Title: Using real-time UV-visible spectrophotometer to assess an Al-Zn-Mg-Cu alloy's dissolution in acidic solution

Manuscript ID: RSOS-200461.R3

It is a pleasure to accept your manuscript in its current form for publication in Royal Society Open Science. The chemistry content of Royal Society Open Science is published in collaboration with the Royal Society of Chemistry.

Yours sincerely,

Dr Laura Smith

Publishing Editor, Journals

Royal Society of Chemistry

Thomas Graham House

Science Park, Milton Road

Cambridge, CB4 0WF

Royal Society Open Science - Chemistry Editorial Office

On behalf of the Subject Editor Professor Anthony Stace and the Associate Editor Dr Dattatray Late.

RSC Associate Editor:
Comments to the Author:
Accept as is

RSC Subject Editor:
Comments to the Author:
(There are no comments.)

Reviewer(s)' Comments to Author:
Reviewer: 1

Comments to the Author(s)
The manuscript is acceptable for publication as is.

Reviewer: 2

Comments to the Author(s)
The addition the authors have put in perfectly addresses the concerns about the assignment of the peaks and should be published with no further changes required.

Appendix A

RSC Subject Editor:

Comments to the Author:

(There are no comments.)

RSC Associate Editor:

Comments to the Author:

1) Introduction need more elaboration highlighting importance of the work and novelty etc.

Reply: Agree. This has been made, see details in our response to Reviewers' comments.

2) XPS data and analysis need to be added.

Reply: To the best knowledge of the authors, we believe that the current data can support the whole story of this work. Thus, if we can have your permission, we will not provide XPS data this time. If you insist, please let us know.

Reviewer comments to Author:

Reviewer: 1

Comments to the Author(s)

This work deserves publication in this journal as it brings an interesting way to investigate corrosion and dissolution of metals. Certainly related strategies have been used or are in use today, some of them referred in this work, however the results are sufficiently different to justify publication in this journal.

Though, the following points must be taken into consideration in the necessary revision.

References 1-5 seem not appropriate to support the great importance of metal dissolution in corrosion or to give further information about the importance of corrosion science and engineering. The criteria for choosing these references, if any, is not clear. It seems that any corrosion paper could have been chosen. Books, Handbooks, Review papers or Critical Perspectives are better references. The previous comment is particularly applicable to references 5, 6 and 7. The references for techniques like polarization, EIS and Noise, should be either dedicated books, review papers or the first papers where each technique has appeared.

Reply: Agree. This has been modified.

Page 2, line 19. The comment that EIS accelerates corrosion is incorrect. This might occur if the technique is not correctly performed, or performed in circumstances where causality, linearity and stability (the three primary conditions that must be met during an EIS experiment for the results be considered valid) are not obeyed. If this

happens it cannot be called an EIS experiment.

Reply: Agree. This has been modified in the revised manuscript. EIS, though three primary conditions that must be met during an EIS experiment can be met, can be seen as a nearly non-destructive method. Strictly speaking, it is destructive because external voltage, though very small, applied on the corrosion process.

Page 3, lines 23-27. It is also incorrect to say that microscale and nanoscale techniques are destructive. They just follow the natural corrosion process, which is by itself destructive due to its very nature. The techniques do not interfere with the process. The same applies to the UV-Vis spectrometry technique presented in this work.

Reply: Thank you! Modifications have been made regarding this comment. However, not every micro/nano-scale technique is non-destructive. For example, whether SECM is destructive depends on its mode: non-destructive for substrate generation-tip collection mode, while for feedback mode it is destructive because the species generated from the tip contributes to the corrosion process of the substrate.

Page 3, lines 30-31. These lines make no sense. The claim that non-destructive methods need to be developed implies that they do not exist. They do! Plenty of methods exist, including EIS, SECM, SKPFM, optical monitoring, etc. A clear difference must be stated between the form of inducing degradation (which can be natural or can be forced, with polarization for example) and the methods to analyse the degradation (which can be destructive or non-destructive). In any case, the natural process of corrosion is destructive. The approach proposed by the authors is a spectroscopic method (non-destructive) with the dissolution of an alloy in acid (destructive). This must be clarified to avoid confusion and misinterpretations.

Reply: Agree. It has been clarified. E.g.,

*In order to better understand corrosion kinetics under natural condition without any applied potential, non-destructive methods **that do not interfere with the natural corrosion process have been developed.***

Page 2, last line. The 10 minutes are presented here as a limitation. No. If necessary the measurements can be continued for any time desired.

Reply: The related sentences have been deleted according to the comment.

Page 3, line 15. Which debate are authors referring to?

Reply: “debate” has been replaced with “issue”.

2.1. Materials

What is T3 Cu?

From where did authors obtained the Al binary alloys? Sizes and shapes? The compositions were uniform and homogeneous along all the volume of the samples?

Reply: Because it has a high enough purity to determine the Uv/vis response of Cu species. The size and state of Al binary alloys have been added.

2.2. STEM

What was the main voltage used with the field emission gun?

Reply: It varies from 2 to 30 kV, and the final voltage is 5 kV.

The acronym HAADF must be introduced.

Reply: Have been added

2.3. UV-vis

Line 49. Revise "...the solution was carried out in a (...) glass vessel..."

Reply: Thank you. This has been revised.

The caption of Fig 1 does not need to state the experimental details that are already written in the experimental section.

Page 4, line 49. It is referred that mass loss tests were performed, but there is no information about it in the experimental section. Measurements after what time of immersion?

Reply: These have been deleted and done.

Page 6, line 44. Introduce the apparatus used for the PVM.

Reply: Has been added.

Table 1. Include in the caption the time of immersion.

Reply: Has been added.

Page 10, line 22. The CPE is related to the passive films, as is written, or to the double layer capacitance?

Reply: Should be due to double layer capacitance. Thanks

Figure 6. The Nyquist plot in a) must have orthonormal axes (same length for the same impedance in both axes).

The legend of Y axis in c) is strange. Why the minus signal? And it should be "log |Z|".

Reply: Has been modified accordingly. Thank you!

Page 11, first line. "fake" should be replaced by a better word.

Reply: "fake" has been deleted.

Figure 7. It is better to put axes in the graph. The colour legend indicates “paicles” instead of particles. Consider changing the colour dots in the image: blue for solution and grey for the alloy. Seems more common-sense.

Reply: This has been modified.

Conclusions

The “non-destructive” before UV-Vis spectrophotometer is unnecessary. Because it is obvious the technique does not interfere with the metal degradation, and also because it may lead to a wrong conclusion: the metal corrodes and is destroyed in a natural manner but the way it is written it seems that at the end of the experiment the sample remains pristine, unspoiled.

Reply: This has been deleted.

“due to the empty d orbital of Cu^{2+} ” is not a conclusion from this work. It should be deleted without any prejudice to the message.

Reply: This has been deleted.

Review the last 4 words, “complexes that spectrum detectable.”

Reply: This has been modified.

References 6 and 17 are the same.

Reply: This has been modified.

Reviewer: 2

Comments to the Author(s)

While the manuscript is interesting and describes the possible uses and limitations of UV-vis for in-situ metal dissolution, there is one main question issue that needs to be clarified and some minor comments.

While very well written and easy to understand there are a few minor errors that could be fixed with a proof read.

In the introduction there is a sentence : Mass loss is a good and conventional..... The word good is very vague and should just be removed, so it reads something like: Mass loss is a conventional method for corrosion rate.....that is relatively easy to conduct.

Reply: This has been modified accordingly: “*Mass loss is a conventional method for corrosion rate assessment that is relatively easy to conduct.*”

Also in the introduction it says that polarization polarises to several hundred millivolts. Is the point here that polarisation is destructive or that it is not a reliable measure of corrosion current? While it is destructive, the reliability of a polarisation test should always be confirmed by conducting separate tests for the anodic and cathodic arm that should match the full scan. If this is done then it is a reliable method.

Reply: The point is that polarization is destructive, which would interfere the corrosion process.

Need references for the statement that EIS accelerates corrosion p2, line 19.

Reply: The sentence stated that EIS accelerates corrosion has been deleted, according to another reviewer's comment.

Final line of the introduction states that: It was of interest to apply this real-time non-destructive technique to the dissolution of Al alloy and see if new insights into this debate might be forthcoming. A debate means that there are opposing views as to what is occurring, what are they? Or is this technique just making the processes that occur clearer?

Reply: Thank you for the language advice, the word "debate" has been replaced with "issue".

The results section is clear and well written, however there is one major issue that needs to be cleared up. Figure 4a and Table 1 shows that for a similar level of Cu in solution for Al-2Cu (8.3mg/L) and Mg for Al-2Mg (7.5mg/L) the UV-vis shows a roughly 4 times higher Cu signal (intensity 0.092 for Cu to 0.026 for Mg). However, for the AA7150, Table 1 shows around 20 times as much Mg as Cu present in solution, but this is ignored as the author states that the absorbance at 252nm is mainly accounted for by copper. Is the UV-vis intensity not proportional to concentration? Please explain how the signal for Mg in the AA7150 solution can largely be ignored when there is 20 times more in solution and its signal was shown to be roughly 4 times smaller than that of Cu for a comparable amount in solution.

Reply: Thank you for the fantastic comment. We already mentioned Mg's effect on absorbance, as we stated in the original manuscript: "*Also can be found is that Mg-containing complex plays a secondary role on absorbance.*". UV-vis intensity should be proportional to concentration, according to Beer's law, however, the key question is, to concentration of which species? To be honest, at this stage, we don't know the accurate answer of this key question.

If we assume that absorbance is proportional to concentration of Mg, based on 0.026 for 7.5mg/LMg in Al-2Mg, then 61.4 mg/LMg in AA7150 would lead to the absorbance intensity to be 0.21. If we assume that absorbance is proportional to

concentration of Cu, based on 0.092 for 8.3mg/LMg in Al-2Mg, then 3.1 mg/LMg in AA7150 would lead to the absorbance intensity to be 0.034. Then the absorbance, in total, should be 0.244, which is much larger than the measured value (0.102). Therefore, the proportional relationship does not exist for a specific ion species.

The absorbance of pure Cu is very strong. Compared Cu and Al-2Cu, a nearly proportional relation can be found for Cu concentration and absorbance, thus we claimed that the absorbance at 252 nm mainly due to Cu species. One explanation of AA7150 is that, the formation of absorbance sensitive Mg complexes is suppressed in the presence of Cu ions, and probably Cu-Mg-containing complexes formed, but these hypotheses still need further proofs.

Appendix B

Reviewer comments to Author:

Reviewer: 2

Comments to the Author(s)

The authors have addressed most of the reviewers comments well, however a question remains as to how much the Mg in AA7150 contributes to the UV-vis signal. While it is clear that Cu gives a stronger signal than Mg, there is much more Mg present in the alloy, however the authors choose to ignore its contribution. This may be true, but proof is needed either from a reference or through some experiments in which the amounts of Cu and Mg together in a solution are varied and the effect on the UV-vis signal is determined. The authors themselves state: '... the formation of absorbance sensitive Mg complexes is suppressed in the presence of Cu ions, and probably Cu-Mg-containing complexes formed, but these hypotheses still need further proofs.' Since this is the main point of the manuscript then it needs to be proven.

Reply: Thank you my dear reviewer for your great insisting, this really helps. In order to answer this question, we examined the raw data. As shown in Fig.1, five curves of parallel experiments can be seen. Curve #1 is the one presented in manuscript R1 and the original manuscript. I really do not know why I chose #1 to represent Al-2Mg, cause the absorbance of #1 is much higher than the other four curves. Probably I paid almost all of my attention to curves of Cu and Al alloys containing Cu, because the story of Cu was as expected. For a parallel experiments, curve #1 actually should be discarded (probably due to residual solution of last testing sample). Thus, in order to ensure scientific accuracy, curve #1 has been replaced by curve #3 with an average absorbance intensity. Please see Fig. 4 in the manuscript R2.

I don't know whether this reply can answer your concern? This is so embarrassed that I took five parallel experiments but put the highest curve to publish. Mg was largely ignored during the research, data analysis and paper writing, compared with Cu. Because Cu is the one and its story meet my hypothesis.

Thanks again and be safe in such a special period.

Fig. 1 Reproductivity of UV/vis response of Al-2Mg corrosion solution

Fig. 4 in the manuscript R2

Reviewer: 1

Comments to the Author(s)

This reviewer is satisfied with the changes made by the authors.

Just pay attention to the following:

Kelvin is a name so in page 2 line 7 write “scanning Kelvin probe force microscopy”

Reply: This has been modified.

Page 2 lines 9-11. The phrase is more correct if written as “Some of these micro/submicro-scale techniques may sometimes interfere with the natural corrosion process...”.

Contrary to what this introduction tries to state most localised techniques do not interfere with the corrosion process: LEIS, SKP, SKPFM, SVET, even SECM (because in SECM corrosion studies it is pH, O₂ or metal ions reduction - after the natural oxidation by the corrosive process - that are measured, without any external polarization).

Reply: This has been modified.

Give the source/producer of the binary alloys (Al-8Zn, Al-2Mg, Al-2Cu, Al-2Mn).

Reply: These alloys were made by ourselves in our own lab.

Always state the country of each equipment manufacturer.

Reply: This has been added.

EIS. Fig. 6. Only 1 time constant?

Reply: Yes, because only one peak can be observed in Bode-phase plot.

Appendix C

Reviewer: 1

Comments to the Author(s)

Just two small points needing revision.

1) But some of these micro/submicro-scale techniques MAY/MIGHT interfere with natural corrosion process...

2) In the experimental part you should say that the binary alloys were produced by yourselves (and give some small details about how they were prepared).

Reply: Thank you for your suggestions, this has been modified accordingly.

Reviewer: 2

Comments to the Author(s)

While the authors have made a valid change to the data presented that explains why they believe Cu is the major contributor to the UV-Vis peak in the AA7150 solution, they have not changed the peak reading in their discussion and still refer to a peak height for the Al-2Mg solution of 0.026 (p10 line 7) when the new curve shows a value closer to 0.01. Additionally, they seem to have chosen the peak height of just one of their repeats (the number of repeats was not mentioned in the experimental section) and offer no average values or error analysis. While this new value of the peak height is now close to those of Zn and Mn, which the authors state are so low they can be ignored, a more thorough explanation on why these can be ignored should be supplied because, unlike for Mn and Zn, the ICP results in Table 1 show that the amount of Mg present in solution for the AA7150 (61.4 mg/L) is much increased as compared to the Al-2Mg solution (7.5) from which the calibration for Mg is measured, so if there is any contribution from Mg in the UV-Vis peak it will be much increased in the AA7150 measurement. This is also important because the UV-Vis peak height in Figure 4 for the AA7150 solution actually increases compared to that of the Al-2Cu solution, even though Table 1 shows less than half the amount of Cu in solution for the AA7150 solution (3.1 vs 8.3 for the Al-2Cu solution). If the authors can explain in the manuscript why a peak height of around 0.1 on the UV-Vis can be ignored then the paper should be accepted.

Reply: Thank you again, as sharp as previous comments, we appreciate it. Our key response is that if Mg is the main reason, an abrupt turning point would disappear. The curve shown in Fig. 3b and 5b would show a gradual increasing trend, instead of the current one. The discussion and analysis in the manuscript-R3 are as below:

“At the wavelength of 252 nm, strong absorbance peaks are found for AA7150 and Al-2Cu, with intensity of 0.102 and 0.092, respectively, implying the important role of Cu. While for Al-2Mg, Al-8Zn and Al-2Mn, the absorbance peak is much less

pronounced. The absorbance intensity at the wavelength of 252 nm of Al-2Mg, Al-8Zn and Al-2Mn is 0.011, 0.009 and 0.012, respectively, which is very close with each other. For Al-2Mn, a flat peak at 340 nm with absorbance intensity of 0.008 also can be observed, which should be due to Mn-containing complexes. And as expected, the peak at 340 nm is absent for AA7150 due to its low content of Mn. As can be seen from Table 1, the concentration of Al is quite comparable for Al-2Mg, Al-8Zn and Al-2Mn alloys, which is the same with absorbance values at 252 nm. Therefore, it is reasonable to believe that for Al-2Mg, Al-8Zn and Al-2Mn, the absorbance at 252 nm is mainly due to Al-containing complexes. Comparing ions concentrations and UV-vis results of Al-8Zn and AA7150, it is easy to find that the effect of Zn on absorbance is negligible. Readers probably question why the absorbance at 252 nm of Al-2Mg is not mainly due to Mg ion, as for Al-2Mg and AA7150, the Mg concentration increases from 7.5 to 61.4 mg/L and the absorbance increases from 0.011 to 0.102. According to K. Ogle's work ¹³, Mg and Al ions are released into solution immediately during dissolution. Thus, if Mg-containing complexes play a major role in absorbance for Al-2Mg and AA7150, then the absorbance curve depicted in Fig. 3a should have a gradual increasing trend, instead of an abrupt turning point at 7.5 h. Therefore, for Al-2Mg alloy it is Al-containing complexes, instead of Mg species, that mainly accounted on the UV-vis response. We further conducted ex-situ UV-vis measurement for T3 pure Cu, and the presence of ultra-strong absorbance peak at 252 nm shown in Fig. 4b strongly indicates Cu's contribution. In addition, comparing absorbance and concentration information of 7150 Al alloy, Al-2Cu and other Al binary alloys, Cu's contribution on absorbance is much stronger than that of Al. Therefore, based on the above analysis, the absorbance at 252 nm is mainly accounted on copper. Another thing worthy noted is that the concentration of Cu is not simply proportional to absorbance intensity. For instance, after subtracting the contribution of Al (~ 0.01), Cu absorbance of pure Cu, Al-2Cu and AA7150 respectively is 3.4, 0.08 and 0.09, while the respective Cu concentration is 154.1, 8.3 and 3.1 mg/L. This probably is due to the formation and solution structures of Cu-containing complexes are affected by the presence of other species like Al, Mg and Zn ions. ”